# Prevalence and Associated Factors of Successful Ageing among People 50 Years and Older in a National Community Sample in Thailand

**DOI:** 10.3390/ijerph191710705

**Published:** 2022-08-27

**Authors:** Dararatt Anantanasuwong, Supa Pengpid, Karl Peltzer

**Affiliations:** 1Center for Aging Society Research (CASR), National Institute of Development Administration (NIDA), Bangkok 10240, Thailand; 2Department of Health Education and Behavioral Sciences, Faculty of Public Health, Mahidol University, Bangkok 10400, Thailand; 3Department of Public Health, Sefako Makgatho Health Sciences University, Pretoria 0208, South Africa; 4Department of Psychology, University of the Free State, Bloemfontein 9300, South Africa; 5Department of Psychology, College of Medical and Health Science, Asia University, Taichung 413305, Taiwan

**Keywords:** ageing, health, Thailand

## Abstract

The aim of the study was to assess the prevalence and associated factors of successful ageing (SA) among people 50 years and older in Thailand. We analyzed national cross-sectional data (5092 men and women 50 years or older) from the Health, Aging and Retirement in Thailand (HART) study in 2015. The SA measures included (1) life satisfaction, (2) social engagement, (3) no major illness, (4) no probable depression, and (5) absence of functional disability. The sample included 5092 participants (median age 67 years, interquartile range 60 to 78 years). The prevalence of SA was 60.0% in adults 50 years and older, ranging from 43.8% in Krabi province to 80.2% in Pathum Thani province, 58.1% (≥60 years), and 56.3% (≥65 years), and the prevalence of the components of SA was 92.3% without major illness, 96.1% without functional disability, 87.5% without probable depression, 91.3% social engagement, and 82.3% high life satisfaction. In multivariable Poisson regression analysis, Buddhist religion (adjusted Prevalence Ratio (aPR): 1.50, 95% Confidence Interval (CI): 1.25 to 1.79), high subjective economic status (aPR: 1.29, 95% CI: 1.11 to 1.49), and physical activity (≥150 min/week) (aPR: 1.11, 95% CI: 1.01 to 1.24) were positively associated and increasing age (aPR: 0.993, 95% CI: 0.989 to 0.997) was negatively associated with SA. Almost two in three older adults in Thailand were successfully ageing. Factors associated with SA included being Buddhist, younger age, higher subjective economic status, and higher engagement in physical activity. These identified factors should be incorporated into health promotion intervention programs in Thailand.

## 1. Introduction

Thailand is a rapidly ageing society. Life expectancy at 60 years of age increased from 20.2% in 2000 to 21.9% in 2015 [1]. The Thai National Plan for the Elderly considers elderly persons with good living standards as follows: “Physically and mentally healthy; Happy family, social care, enabling and friendly environment; Stable security, access to appropriate welfare and service; Lead a valuable life with dignity, independence and autonomy, and serve as central reliability and participate in the family, community and social activities; Maintain access to data, information, and news” [2]. According to the Active Ageing Index (AAI), which incorporates health, participation, security, and enabling factor indices, in addition to income and health, social engagement, lifelong learning, and work choices contributed to a higher level of AAI in Thailand [3].

Successful ageing (SA) can be defined using a multidimensional concept, including life satisfaction, mental well-being, social engagement, no disability, and no major illness [4]. Using the same definition of SA, in China (≥65 years), the prevalence of SA was 18.6%, in Korea, it was 25.2% [4], in India (≥65 years), it was 27.2% [5], and in South Africa (≥50 years), it was 36.6% [6]. As reviewed previously [5,6], indicators associated with SA comprise sociodemographic indicators (married, male sex, younger age, higher economic status, higher education, ethnicity, and region) and health behaviors, such as physical activity, not smoking, and normal body mass index. Furthermore, diet quality [7] and intellectual and developmental disabilities [8] can influence successful ageing.

Based on a meta-analysis, older adults with SA have a 50% reduced risk of all-cause mortality [9]. Therefore, having national data on SA can provide greater insights in evaluating well-being among older adults in Thailand. We were unable to identify a national study on SA in Thailand [10], which prompted this study. The aim of the study was to assess the prevalence and associated factors of SA among older adults in Thailand.

## 2. Methods

### Sample and Procedure

We analyzed national cross-sectional data from the 2015 Health, Aging and Retirement in Thailand (HART) study. Households (*N* = 5600, from five regions and Bangkok) that had at least one member (≥45 years) were randomly selected using a multistage sampling design (1 = 6 strata of regions, 2 = capital and other district of the province, 3 = villages or blocks, and 4 = one household member). Data were collected by trained interviewers from February to July, 2015. Full details of the sampling procedures have been published previously [11].

In HART 2015, a baseline sample of 5616 men and women aged 45 years or older was interviewed. The study was conducted by trained field workers at the participants’ homes using a paper-and-pencil (PAPI) questionnaire. We restricted our analytical sample to those 50 years and older (*N* = 5092). The study received ethical approval from the Ethics Committee in Human Research, National Institute of Development Administration—ECNIDA (ECNIDA 2020/00012). Participants provided written informed consent prior to the study.

## 3. Measures

*SA* was assessed using a multidimensional concept, including life satisfaction, social engagement, no major illness, no functional disability, and no probable depression [4].

Absence of major illnesses was assessed with self-reported healthcare provider-diagnosed cardiovascular diseases, heart disease, heart failure, brain diseases/Alzheimer’s disease, lung diseases/emphysema, and cancer.

No functional disability was measured based on a modified Activities of Daily Living (ADL) index [12], which asked respondents if they needed help with four ADLs (eating, bathing, dressing, and washing). The response options ranged from 0 = “able to do it all by myself” to 3 = “need help for all steps”. The absence of functional disability was defined as a respondent being able to perform all four items by themselves. The reliability coefficient of the ADL scale was Cronbach’s α = 0.94 in this study.

Free of probable depression (<10 scores) was assessed with the Center for Epidemiologic Studies Depression (CES-D-10) scale [13]. In a previous study among adults in Thailand, scores ≥10 showed “sensitivity of 96.7% and specificity of 86.6% for depression” [14]. The CES-D has also been found valid for use in older community samples, including Thailand [15]. The CES-D10 had a reliability coefficient of 0.78.

Social engagement included formal and informal social engagement. Formal social engagement (defined as at least one activity) was measured with six items: religious, occupational, and cultural organizations; alumni or parent association or association of people from the same hometown, volunteering, and political organizations [16]. Responses were coded as “1 = daily to at least once a month” and “0 = once a year or never” [16] (Cronbach’s alpha was 0.7). Informal social engagement was determined with two items: (1) “In the past year, do you have any close friends or relatives who live nearby and have a close relationship with? (Please refer to the only person whom you meet most often)”, and (2) “If so, how often do you meet with them in person (number of times per day, week, month, year, other, never)?” Informal social engagement was defined as “1 = having a close friend or relative who lives nearby and have a close relationship with and having met that person at least in the past one month”, and “0 = not having a close friend or relative or meeting a close friend less than once a month in the past year”. Positive responses were summarized for questions related to formal or informal social engagements.

Life satisfaction was assessed with the question, “In overall, how satisfied are you with your quality of life (or how happy do you feel)?” Quality of life or happiness was rated from 0 to 100, with 100 indicating the highest quality of life or happiness. Life satisfaction was defined as 70 to 100.

### 3.1. Covariates

Sociodemographic information included age, education, sex, marital status, religion, and income quartile. The income quartile was calculated based on annual income from employment, own business, agricultural/livestock/fishing business, short-term or contract work, financial support from family, renumeration/pension income from a government fund, occupational pension fund, private pension fund, social security/welfare income, income from government living allowance, veteran’s welfare benefit, other welfare assistance income, and income from other sources, and divided into four groups: 1 = 0 to <13,000 Thai Baht, 2 = 13,000 to <50,000, 3 = 50,000 to <140,000, 4 = ≥140,000 Thai Baht (average exchange rate in 2015: 1 US$ = 34.2 Baht) [17].

Subjective economic status was assessed with the item “How satisfied are you with your economic status?” Responses ranged from “0” to “100”, and were grouped into low = 0–40, medium = 50–70, and high = 80–100.

Tobacco smoking was assessed with the question, “Have you ever smoked cigarettes?” (response options: 1 = yes, and still smoke now, 2 = yes, but quit smoking, and 3 = never).

Heavy alcohol use was defined as having 3 and 2 or more units of alcoholic beverages in one session in the past month, for men and women, respectively.

Physical activity was assessed with the frequency and duration of any type of exercise in the past week [18], and categorized as none = inactivity, 1–149 min/week = low activity, and ≥150 min/week = high activity [19].

Body mass index (BMI) was assessed via self-reported body weight and height and classified into underweight (<18.5 kg/m^2^), normal weight (18.5–22.9 kg/m^2^), overweight (23–24.9 kg/m^2^), and obesity (25+ kg/m^2^) [20].

### 3.2. Data Analysis

All statistical analyses were performed with StataSE 15.0 (College Station, TX, USA). The frequency distribution of SA and its components was calculated. Univariable and multivariable analysis was conducted using Poisson regression to estimative prevalence ratios (PRs) and confidence intervals (95% CI). Significant variables in univariable analyses were subsequently included in multivariable analyses. Income and subjective economic status were separately analyzed in the multivariable models. *p*-values < 0.05 were considered significant. The variance inflation factor (VIF) was calculated to check for multicollinearity, and none was found between the study variables.

## 4. Results

### 4.1. Sample Characteristics

The sample included 5092 participants (≥50 years, median age 67 years, interquartile range 60 to 78 years). The prevalence of female participants was 52.3%, 93.0% were Buddhist, 4.9% had heavy alcohol use, 11.7% were current smokers, 15.4% had high physical activity, and 30.7% had obesity. The prevalence of SA was 60.0% in adults 50 years and older, 58.1% (≥60 years), and 56.3% (≥65 years), and the prevalence of the components of SA was 96.1% without functional disability, 92.3% without major illness, 91.3% social engagement, 87.5% without probable depression, and 82.3% with high life satisfaction. The sample details are shown in Table 1.

The prevalence of SA differed by province, from 43.8% in Krabi province to 80.2% in Pathum Thani province. Analyzing different age groups, the prevalence of the four components of SA (no functional disability, no major disease, no probable depression, and social engagement) decreased significantly from 50–64-year-olds to those 85 years and older, while the prevalence of high life satisfaction remained unchanged across the age groups (see Table 2).

### 4.2. Associations with SA

In univariable Poisson regression analysis, Buddhist religion, upper middle and higher income, medium and high subjective economic status, and physical activity were positively associated and increased age was negatively associated with SA. In multivariable Poisson regression analysis, being Buddhist (adjusted Prevalence Ratio (aPR): 1.50, 95% Confidence Interval (CI): 1.25 to 1.79), high subjective economic status (aPR: 1.29, 95% CI: 1.11 to 1.49), and physical activity (≥150 min/week) (aPR: 1.11, 95% CI: 1.01 to 1.24) were positively associated and increasing age (aPR: 0.993, 95% CI: 0.989 to 0.997) was negatively associated with SA (see Table 3).

## 5. Discussion

This study aimed, for the first time, to provide data on SA among older adults (≥50 years) in a national community-based sample in Thailand in 2015. Using a multidimensional concept of SA, we found that almost two in three older adults (≥50 years) (60.0%) in Thailand were successfully ageing; the figures were slightly lower for those ≥60 years (58.1%) and ≥65 years (56.3%). Thailand appears to have a prevalence of SA that is higher than in China (*N* = 15,191; ≥65 years, 18.6%), Korea (*N* = 4155; ≥65 years, 25.2%) [4], India (*N* = 21,343; ≥65 years, 27.2%) [5], and South Africa (*N* = 3734; ≥50 years, 36.6%) [6]. The SA component “no major illness” (≥50 years 92.3%; ≥65 years 90.0%) in this study was higher than in China (≥65 years, 75.1% [4], India (≥65 years; 83.3%) [5], and South Africa (≥50 years; 73.3%) [6]. Similarly, the prevalence of the SA component “without disability” in Thailand was higher than in China, Korea, India, and South Africa. These differences in results may be related to the different definitions used; for example, some studies included instrumental activities of daily living, while our study only included activities of daily living, and it may also be that Thai older adults are less aware of their major illnesses than in China [4]. The prevalence of no depression (87.5%) in this survey was higher than in China (75.2%) [4] and lower than in India (91.8%) [5], while the prevalence of social engagement (91.3%) and high life satisfaction (82.3%) in this study was higher than in China (51.2% and 57.1%, respectively) [4] and India (73.6% and 74.6%, respectively) [5]. Again, some of these country differences may be influenced by different classifications; for example, this study only measured depressive symptoms and the Indian study assessed major depressive disorder [5], and, in terms of social engagement, the China study [4] included productive engagement, and this study included informal social engagement.

Among the different components of SA by age group, the decline with age was strongest for no disability and to a lesser extent for no disease, no probable depression, and social engagement, while life satisfaction remained unchanged. These findings are consistent with a study on older adults in India [5]. The highest prevalence of SA was found in the Thai provinces of Pathum Thani, Khon Kaen, Surin, Chanthaburi, and Samutprakarn, and the lowest was found in Krabi, Songkhla, Phetchabun and Nonthaburi. These differences by province may be explained by diverging levels of economic development and life expectancy. For example, the Gross Provincial Product (GPP) per capita for Krabi in 2015 was B187,258 or US$5467.39 (B34.2498 = US$1.00) (in 2020 = B163,070 or US$5212.94 (B31.2818 = US$1.00)), which was lower than that of Pathum Thani (in 2015, GPP per capita was B219,440 or US$6407.04; in 2020, B239,753 =US$7664.31) [21].

We found that factors associated with SA included being Buddhist, younger age, higher subjective economic status, and greater participation in physical activity. The inverse relationship between SA and age relates to functional, biological, and cognitive decline with increasing age [22]. Consistent with previous research [5,23], we found that a higher subjective economic status increased the odds of SA in this investigation. However, higher income and higher education were not significantly associated in the adjusted model with SA in this study. Older adults with better economic status may make use of resources enabling them to engage in increased healthy behaviors [24]. Furthermore, we found differences in SA according to religion. Compared to Muslims and Christians, Buddhists had higher odds of SA in this survey. Muslims and Christians had a particularly lower rate of the SA components of no functional disability, no probable depression, and life satisfaction than Buddhists. Similar to a previous study among married women in urban Thailand [25], this study showed that Buddhist participants reported a significantly lower prevalence of probable depression than non-Buddhists. This finding may be explained by the minority status and stress of non-Buddhists, Muslims, and Christians in Thailand. We found that males had a higher prevalence of SA than women, but this was not significant, unlike some previous research [5,26].

Consistent with other studies [5,27], we found that higher participation in physical activity increased the odds of SA. Research has provided evidence [27,28,29] that physical activity improves physical and mental health, including life satisfaction, thus contributing to better SA. Contrary to some former studies [4,30,31], we found no significant association between non-smoking, alcohol use, underweight, overweight/obesity, and SA.

## 6. Study Limitations

The investigation was limited by the assessment of variables using self-reports and the design of the cross-sectional study. Some factors, such as diet pattern [7,32] and cognition, were not measured and should be included in future studies. In addition, the survey excluded institutionalized older adults.

## 7. Conclusions

Almost two in three older adults in Thailand were successfully ageing. Factors associated with SA included being Buddhist, younger age, higher subjective economic status, and higher engagement in physical activity. These identified factors should be incorporated into health promotion intervention programs in Thailand.

## Figures and Tables

**Table 1 ijerph-19-10705-t001:** Sample and successful ageing characteristics in older adults (≥50 years) in Thailand, 2015.

Variables	Subcategory	Sample	No Major Illness	No Functional Disability	No Probable Depression	Social Engagement	Life Satisfaction	Successful Ageing
Age (in years)	Median (interquartile range)	67 (18)	67 (18)	67 (18)	67 (18)	67 (18)	68 (18)	66 (17)
		*N* (%)	%	%	%	%	%	%
	65 years and older	3011	90.0	93.9	86.1	90.0	82.6	56.3
	60 years and older	3484	91.0	94.9	86.6	90.6	82.5	58.1
All	50 years and older	5092	92.3	96.1	87.5	91.3	82.3	60.0
Sex	Female	2663 (52.3)	92.5	96.2	86.3	91.8	81.3	58.8
	Male	2429 (47.7)	92.1	95.9	88.7	90.8	83.4	61.3
Education	≤Elementary	4285 (84.5)	92.1	95.7	86.7	91.9	82.0	59.3
	>Elementary	788 (15.5)	93.4	97.9	91.4	88.6	83.5	63.2
Marital status	Not married	2217 (43.6)	91.7	94.6	86.5	90.8	82.5	58.2
	Married/cohabiting	2870 (56.4)	92.8	97.2	88.2	91.7	82.1	61.4
Religion	Muslim or other	354 (7.0)	92.4	91.4	67.4	96.6	72.3	40.2
	Buddhist	4732 (93.0)	92.3	96.4	89.0	90.9	83.0	61.5
Income quartile	Low	1326 (26.0)	89.3	95.4	82.4	91.1	83.3	55.5
	Lower middle	1338 (26.3)	91.0	92.8	85.8	91.7	82.4	56.6
	Upper middle	1260 (24.7)	93.4	97.5	90.5	91.3	81.4	62.3
	High	1168 (22.9)	96.0	99.0	91.7	91.1	82.0	66.3
Subjective economic status	Low	461 (9.4)	88.7	94.7	79.4	86.8	82.0	49.8
	Medium	2855 (58.1)	91.8	95.7	87.3	90.2	82.6	58.7
	High	1595 (32.5)	94.4	97.3	91.2	94.7	82.1	66.0
Heavy alcohol use	No	4843 (95.1)	92.1	95.9	87.3	91.2	82.2	59.6
	Yes	249 (4.9)	96.0	98.4	90.9	94.0	83.5	68.1
Smoking tobacco	Never	4087 (80.3)	92.8	95.9	87.2	90.9	82.2	59.7
	Past	410 (8.1)	84.1	92.8	86.8	92.2	85.6	54.6
	Current	595 (11.7)	94.8	99.5	89.5	93.8	80.5	65.3
Physical activity	None	3066 (60.2)	91.9	94.3	85.2	90.5	82.2	56.9
	1–149 min/week	1242 (24.4)	93.0	98.1	89.9	91.7	82.9	63.8
	≥150 min/week	784 (15.4)	92.6	99.5	92.3	94.0	81.6	65.5
Body mass index	Normal	1744 (38.1)	92.5	95.6	87.3	92.8	82.4	60.6
	Underweight	522 (11.4)	89.7	92.4	83.6	91.4	84.7	55.5
	Overweight	907 (19.8)	93.4	97.9	89.8	91.0	81.7	62.2
	Obesity	1407 (30.7)	92.7	97.8	89.1	91.8	83.4	62.6

**Table 2 ijerph-19-10705-t002:** Sample and successful ageing characteristics in older adults (≥50 years) in Thailand, 2015, by province and age group.

Variables	Sample	No Major Illness	No Functional Disability	No Probable Depression	Social Engagement	Life Satisfaction	Successful Ageing
	*N* (%)	%	%	%	%	%	%
Province							
Bangkok	546 (10.7)	87.5	97.0	91.0	83.3	88.3	56.3
Samutprakarn	179 (3.5)	85.5	96.1	97.2	93.9	84.9	65.5
Nonthaburi	189 (3.7)	97.9	95.2	91.3	63.5	88.9	52.5
Pathum Thani	180 (3.5)	93.3	96.1	92.1	97.2	96.1	80.2
Sing Buri	365 (7.2)	88.5	97.8	92.4	87.1	91.2	64.1
Chanthaburi	351 (6.9)	92.0	99.1	88.0	94.6	84.6	67.9
Surin	567 (11.9)	95.9	98.8	91.4	95.2	79.9	68.4
Khon Kaen	452 (10.6)	94.4	98.0	91.9	92.9	86.2	69.5
Chiang Mai	542 (10.6)	93.9	96.3	89.2	97.2	79.5	62.7
Uttaradit	370 (7.3)	93.5	96.4	93.3	92.1	88.1	68.4
Phetchabun	588 (11.5)	95.2	98.1	86.5	91.5	67.3	49.2
Krabi	364 (7.1)	88.2	97.5	78.9	98.1	67.3	43.8
Songkhla	513 (10.1)	92.6	84.0	65.0	91.8	83.4	45.1
Age group in years							
50–64	2081 (40.9)	95.6	99.1	89.4	93.2	81.9	65.3
65–74	1370 (26.9)	91.5	97.7	87.9	91.6	82.6	60.8
75–84	1197 (23.5)	88.2	93.9	85.9	89.1	83.3	54.5
85 or more	444 (8.7)	90.3	82.8	81.1	87.6	80.4	47.4
*p*-value		<0.001	<0.001	<0.001	<0.001	0.542	<0.001

**Table 3 ijerph-19-10705-t003:** Prevalence ratios for the associations between sociodemographic factors, health factors, and successful aging, HART 2015.

Variables	Subcategory	CPR (95% CI)		APR (95% CI)	
Age (in years)	Scale	1 (Reference)		1 (Reference)	
		0.991 (0.988 to 0.995)	<0.001	0.993 (0.989 to 0.997)	<0.001
Sex	Female	1 (Reference)		-	
	Male	1.04 (0.97 to 1.12)	0.257		
Education	≤Elementary	1 (Reference)			
	>Elementary	1.07 (0.96 to 1.18)	0.213	-	
Marital status	Not married	1 (Reference)		-	
	Married/cohabiting	1.05 (0.98 to 1.14)	0.165		
Religion	Muslim or other	1 (Reference)		1 (Reference)	
	Buddhist	1.53 (1.28 to 1.82)	<0.001	1.50 (1.25 to 1.79)	<0.001
Income quartile	Low	1 (Reference)		1 (Reference)	
	Lower middle	1.03 (0.93 to 1.14)	0.712	1.03 (0.93 to 1.15)	0.543
	Upper middle	1.12 (1.01 to 1.25)	0.013	1.06 (0.95 to 1.18)	0.317
	High	1.19 (1.07 to 1.33)	<0.001	1.10 (0.98 to 1.23)	0.112
Subjective economic status	Low	1 (Reference)		1 (Reference)	
	Medium	1.18 (1.02 to 1.36)	0.024	1.18 (1.02 to 1.36)	0.024
	High	1.33 (1.14 to 1.54)	<0.001	1.29 (1.11 to 1.49)	<0.001
Heavy alcohol use	No	1 (Reference)		-	
	Yes	1.14 (0.97 to 1.34)	0.103		
Smoking tobacco	Never	1 (Reference)		-	
	Past	0.91 (0.79 to 1.06)	0.223		
	Current	1.09 (0.98 to 1.22)	0.122		
Physical activity	None	1 (Reference)		1 (Reference)	
	1–149 min/week	1.12 (1.03 to 1.22)	0.011	1.10 (1.01 to 1.21)	0.034
	≥150 min/week	1.15 (1.04 to 1.27)	0.007	1.11 (1.01 to 1.24)	0.039
Body mass index	Normal	1 (Reference)		-	
	Underweight	0.92 (0.80 to 1.05)	0.206		
	Overweight	1.03 (0.92 to 1.14)	0.629		
	Obesity	1.03 (0.94 to 1.13)	0.486		

CPR = Crude Prevalence Ratio; APR = Adjusted Prevalence Ratio; CI = Confidence Interval.

## Data Availability

Data is publicly available at Gateway to Global Ageing Data, Health, Aging, and Retirement in Thailand: https://g2aging.org/?section=study&studyid=44 (accessed on 2 July 2022).

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
