# Peer review of "Prevalence and Associated Factors of Successful Ageing among People 50 Years and Older in a National Community Sample in Thailand"

_ijerph, 2022, doi:10.3390/ijerph191710705_

Round 1
Reviewer 1 Report
The article “Prevalence and associated factors of successful ageing among people 50 years and older in a national community sample in Thailand” by Dararatt Anantanasuwong and co-authors described the factors associated with successfully ageing (SA) among the older adults in Thailand. The present manuscript does not meet the standard of IJERPH. The article needs significant improvement to meet the quality.
(1) Diagram and statistical plots must be incorporated in the introduction, results and discussion section for better understanding.
(2) The results section needs a significant improvement, indigent data representation. However, some of the statistical comparisons can be transferred to the result section from the discussion part.
(3) Introduction, second paragraph, line “The components of SA were in China……. and high well-being 64.0% (Pengpid & Peltzer, 2021a).” quite irrelevant and can be transferred to the discussion section.
(4) Besides the reported factors, what are the roles of food habits/diet and preexisting health conditions/health conditions by birth?
(5) Measures; the second paragraph, please explain “Cronbach’s α = .94”.
(6) In the discussion section, the data comparison did not represent the participant number from each country. Therefore, does the data comparison represent a nearly similar number of participants and gender categories? If not, then how to explain the difference?
(7) Please indicate the time duration involved in this study for 5,092 participants.
Author Response
as attached

Reviewer 2 Report
Nice manuscript. Just a few minor editorial recommendations, please see attached document.

Author Response
as attached

Round 2
Reviewer 1 Report
Please accept the changes. The revised manuscript can be accepted for publication in IJERPH.